# Tube–Iris Distance and Corneal Endothelial Cell Damage Following Ahmed Glaucoma Valve Implantation

**DOI:** 10.3390/jcm11175057

**Published:** 2022-08-28

**Authors:** Yitak Kim, Won Jeong Cho, Jung Dong Kim, Hyuna Cho, Hyoung Won Bae, Chan Yun Kim, Wungrak Choi

**Affiliations:** 1Severance Hospital, Yonsei University College of Medicine, Seoul 03722, Korea; 2Department of Ophthalmology, Institute of Vision Research, Yonsei University College of Medicine, Seoul 03722, Korea

**Keywords:** glaucoma, endothelial cell density, Ahmed glaucoma valve

## Abstract

The most significant factor for endothelial cell loss should be readily identified, since prevention is the most crucial treatment. Here, we investigate risk factors for corneal endothelial cell density (ECD) decline following Ahmed glaucoma valve (AGV) implantation and determine the optimal cut-off values. This study included 103 eyes (95 patients) with glaucoma that underwent AGV implantation between January 2006 and January 2021 at a single medical center (Severance Hospital). We conducted consecutive *t*-tests between two groups separated by the ECD change rate to determine the survival state of the enrolled patients. Associations were evaluated using univariable and multivariable linear regressions. Optimal cut-off values for identified risk factors were analyzed using a Cox proportional hazards model and a receiver operating characteristic (ROC) curve based on logistic regression. Mean follow-up duration was 4.09 ± 2.20 years. After implementing consecutive *t*-tests, only patients with an ECD change rate greater than −6.1%/year were considered to have survived. Tube–iris distance (TID) was the only statistically significant factor identified in both the univariable and multivariable linear regressions. The cut-off value determined from the consecutive Cox regression method was 0.33 mm (smallest *p*-value of 0.0087), and the cut-off value determined from the ROC method was 0.371 mm (area under the receiver operating characteristic curve [AUC], 0.662). Patients with short TIDs showed a better ECD prognosis following AGV surgery; we suggest optimal TID cut-off values of 0.33 mm and 0.371 mm based on the implemented Cox regression and ROC methodology, respectively.

## 1. Introduction

Glaucoma drainage device (GDD) implantation is the preferred therapeutic choice for recalcitrant glaucoma patients who fail antiglaucoma medications or trabeculectomies [1,2,3]. The landmark Tube Versus Trabeculectomy Study demonstrated the superiority of GDD surgery over trabeculectomy with mitomycin C in terms of low failure and revision rates, while the ability to control intraocular pressure (IOP) as well as the number of necessary medications were comparable [2]. Consequently, the use of GDD surgeries has increased over the past two decades, and that of trabeculectomies with mitomycin C has decreased [3].

Endothelial cell density (ECD) damage is a long-term complication of GDD surgery observed at high rates in glaucoma patients [4,5,6,7]. The risks of consequent corneal edema and decompensation due to GDD implantation were twice as high as those associated with trabeculectomy, emphasizing the importance of identifying mediating mechanisms and preventing ECD damage [2,8,9].

Efforts with regard to ECD preservation include GDD insertion into the pars plana (PP) or ciliary sulcus (CS), in contrast to the conventional method of inserting it into the anterior chamber [10,11,12,13]. However, PP insertion requires vitrectomy and can cause additional complications in the posterior chamber [14]. Implantation in the CS also presents few limitations, including difficulty in localizing the tube during surgery (which requires the presence of pseudophakic or aphakic eyes) and the risk of incurring iris pigmentation due to friction [15].

Because of these constraints, most GDDs are inserted into the anterior chamber. Many studies have examined the optimal angle, depth, or location of tube insertion leading to the best surgical outcome [7,16,17,18]. Mendrinos et al. inspected the effects of intracameral tube length (TL), tube–corneal distance (TCD), and tube–iris distance (TID) on peripheral ECD reduction and did not find any significant correlations [7]. In contrast, later studies have consistently detected an association between a longer TCD and lower ECD change rates [16,17]. However, a recent study presented a strong association with the tube–corneal angle (TCA) but not with the TCD [18]. Another study did not target the change in ECD over time but instead determined that the existence of peripheral anterior synechiae (PAS) was associated with a lower ECD at a fixed time point [19].

Although many studies have attempted to discover the mechanisms mediating changes in ECD, to date, no research has comprehensively evaluated all tube and ocular parameters. Hence, we conducted the current study, which evaluated TL, TCA, TCD, TID, insertion–iris distance (IID), and other ocular parameters (such as PAS and angle parameters) and determined the optimal cut-off values for the identified risk factors when implanting an Ahmed glaucoma valve (AGV).

## 2. Materials and Methods

### 2.1. Study Population

This single-center retrospective case series study was approved by the Institutional Review Board of Severance Hospital (Seoul, Korea; IRB No. 4-2022-0043) and conducted in accordance with the requirements of the Declaration of Helsinki. The requirement for informed consent was waived due to its retrospective nature.

Patients with recalcitrant glaucoma unresponsive to maximal medical doses and who underwent AGV implantation (Model FP-7, New World Medical, Inc., Rancho Cucamonga, CA, USA) were reviewed from a prospectively maintained database at the Severance Hospital. A total of 692 patients underwent AGV implantation between January 2006 and January 2021 and at least one examination via specular microscopy before and after surgery. We excluded patients with a history of Descemet membrane endothelial keratoplasty, penetrating keratoplasty or any other baseline corneal disease affecting the corneal endothelium, cataract surgery, or any kind of vitrectomy. We also excluded patients in whom the AGV was not implanted into the anterior chamber (e.g., sulcus, pars plana). A total of 103 eyes were eligible.

These patients were followed up for more than 1 year. All patients had specular microscopy and anterior segment optical coherence tomography (AS-OCT) imaging findings of adequate quality. If additional intraocular surgery, including repositioning or removal of the valve, was required, follow-up was stopped to exclude possible effects on the corneal endothelium from the analysis.

### 2.2. Surgical Techniques

All surgeries were performed by glaucoma specialists. The tip of the AGV was placed in the anterior chamber, following its implantation which was performed as previously reported [20]. Briefly, a fornix-based conjunctival incision was made under sub-Tenon anesthesia, and the Tenon capsule was dissected using spring scissors. Two flaps, a 4 × 4 mm right-angled triangular-shaped partial-thickness scleral flap and a continuous 2 mm wide × 6 mm long bridge-shaped partial-thickness scleral flap, were constructed at the superotemporal or superonasal quadrant. Tube priming was performed using balanced salt solution irrigation, and the AGV body was placed 8–10 mm posterior to the limbus (between the rectus muscles). The tube tip was cut to an adequate length and placed in a bevel-up manner. Viscoelastic was injected to maintain the anterior chamber depth before tube insertion. Using a 23-gauge needle, a sclerotomy was created 1–2 mm posterior to the limbus under the scleral flap, entering the anterior chamber where the tube was inserted parallel to the iris plane. The scleral flap was adjusted over the tube and sutured using a 10–0 nylon suture, and the conjunctiva and Tenon’s capsule were then secured at the limbus with interrupted 8–0 Vicryl sutures. Topical steroids and antibiotic eye drops were prescribed for eight weeks.

### 2.3. Examinations

Medical records were reviewed for preoperative clinical factors, including age at surgery, sex, laterality of the operated eye, past medical history (including hypertension [HTN], diabetes mellitus [DM], tuberculosis [TB], hepatitis, and cerebrovascular accidents [CVA]), glaucoma diagnosis, axial length (AXL), anterior chamber depth (ACD), IOP, postoperative data, and uveitis incidence. Other postoperative data, including ECD and ocular and tube parameters, were obtained as follows. Specular microscopic examination was performed by experienced examiners using a noncontact specular microscope (Topcon SP-3000P; Topcon Corp., Tokyo, Japan). The central area of the cornea was imaged while the patient gazed at the target. The manual center-dot method was used to evaluate central corneal ECD, marking at least 50 contiguous endothelial cells.

### 2.4. Anterior Segment Optical Coherence Tomography (AS-OCT)

The AS-OCT image (Casia SS-1000; Tomey, Nagoya, Japan) closest in time after the operation was used to measure ocular and tube parameters. Because the cross-sectional plane including the tube is not always parallel to the radial section of the eye, variables were evaluated in different cross-sections. 

The TCD and TID were measured when the plane was positioned to involve the tube tip. TCD is defined as the distance between the anterior tip of the tube and the cornea (perpendicular to it). TID is defined as the distance between the posterior tip of the tube and the iris (perpendicular to it). The TCA and the IID were measured when the plane was placed to involve the insertion of the tube. The TCA is defined as the angle between the anterior surface of the tube and the posterior corneal surface. The IID is defined as the distance between the anterior insertion site of the tube and the iris (perpendicular to it). The intracameral length of the tube was not measured directly due to concerns regarding cross-sections and was instead computed by calculating the TCD/tan (TCA) ratio, assuming a right triangle (where TCD represents the length of the opposite side and TL represents the length of the hypotenuse). TCA was measured in degrees and then converted to a radian scale before applying tangent functions. The hypothesis is that the curvature of the posterior corneal surface in this triangle can be neglected (Figure 1). We also measured a range of other parameters, including the angle opening distance (AOD), angle recess area (ARA), trabecular iris space area (TISA), trabecular iris angle (TIA) from both the temporal and nasal sides, iris trabecular meshwork contact (ITC), and anterior chamber width (ACW). The central corneal thickness (CCT) was also determined by AS-OCT since the examination results of some patients were omitted from their medical records.

### 2.5. Statistical Analyses

Python (version 3.8.8 ; Fredericksburg, VA, USA) and statistical software package R (version 3.6.2; The R Project for Statistical Computing, Vienna, Austria) were employed for data manipulation, visualization, and conducting linear regressions. 

Linear regression evaluating the association between ECD and time (years) was performed to derive the coefficient (i.e., the slope of ECD change, cells/mm^2^ × year) for the examined patients. This slope was then divided by the preoperative ECD to derive the ECD change rate (%/year). If a patient had only two data points, the crude slope between them was used instead. ECD change rates were then separated into two groups using a single cut-off value, and *t*-tests were conducted to compare the means. The optimal cut-off value (i.e., the value associated with the smallest *p*-value) was determined from these *t*-tests. Groups A and B were defined as patients with small ECD change rates (less than 6.1% decrease) and with large ECD change rates (more than 6.1% decrease), respectively.

Continuous variables are presented as means ± standard deviations and categorical variables as numbers (percentages). We also compared baseline characteristics between Groups A and B using formal statistical tests. More specifically, continuous variables were subjected to Levene’s test to examine the assumption of variance equality, and the associated *p*-values were derived using Student’s *t*-test or Welch’s *t*-test. Categorical variables were compared using either the chi-square test or Fisher’s exact test. The tube parameters were compared in the same manner.

Univariable linear regressions evaluating associations between the ECD change rate (%/year) and each variable were performed. Selected factors were combined with data on TCD, TCA, and ITC, which were asserted to be significant with regard to ECD change based on previous results [16,17,18,19], in order to conduct a multivariable-adjusted linear regression analysis. A heatmap matrix evaluating Pearson’s r correlations between independent variables was used to interpret collinearity. We prioritized achieving a lean model by excluding variables that showed a high correlation (r > 0.7) [21]. Accordingly, either TCD or TCA were always excluded from multivariable analysis models.

We used two different methods to determine the optimal cut-off value for the TID. First, to predict survival, hazard ratios (HRs, with a referent group defined as TID values less than the cut-off) and *p*-values were compared after running univariable Cox regression analyses over every potential cut-off value between 0.01 mm and 1.25 mm at 0.01 mm intervals (Figure 2A). Event state (y = 1) for this model was defined according to the patient’s categorization into Group B (ECD change rate < −6.1%/year), survival state (y = 0) was defined according to the patient’s categorization into Group A (ECD change rate > −6.1%/year), and follow-up years were used as the time variable.

Second, a receiver operating characteristic (ROC) curve was plotted and the area under the curve (AUC) was then calculated (Figure 2B). The model used to draw the curve was a univariable logistic regression model employing TID as the independent variable. Based on these cut-offs, patients were divided into two groups for constructing Kaplan–Meier (KM) curves, followed by a log-rank test (Figure 3A,B). Finally, we compared postoperative IOP and uveitis incidence between patients with short and long TID using Student’s *t*-test and Fisher’s exact test, respectively. Additionally, other combinations of variables were evaluated in univariable linear regression to support our interpretation of the results.

## 3. Results

### 3.1. Patient Characteristics

Altogether, 103 eyes of 95 patients (mean age, 64.20 ± 13.85 years; 62% male) were included, 54% of the evaluated eyes were right eyes, none of the patients had complications such as tube–corneal touch or collapse of the anterior chamber, and patients were followed up for a mean of 4.09 ± 2.20 years. Primary open-angle glaucoma (POAG) was diagnosed in 55% of the glaucoma patients. The mean preoperative central ECD was 2183.42 ± 527.69 (cells/mm^2^), the mean final ECD was 1765.49 ± 663.50 (cells/mm^2^), and the mean ECD change rate was −4.26 ± 9.49 (%/year). Among 11 cases of neovascular glaucoma, diabetic retinopathy was the primary cause (four proliferative diabetic retinopathies and four non-proliferative diabetic retinopathies), while other three cases were caused by retinal vein occlusions. Additional treatments include panretinal photocoagulation and intravitreal anti-VEGF injections.

### 3.2. Comparison between Patients Separated by Low and High ECD Change Rates

We classified patients into two groups under the assumption that ECD does not decrease in all patients and conducted consecutive *t*-tests to determine the optimal cut-off value regarding the ECD change rate (Appendix A). The *p*-value continuously decreased as the cut-off increased, reaching a minimum point of 7.720 × 10^−16^; the *p*-value continued to increase thereafter. Hence, we considered an ECD change rate of −6.1 (%/year) (i.e., where the *p*-value was the smallest) as the optimal cut-off value, and we thereby separated patients into Group A (with an ECD change rate of >−6.1%/year) and Group B (with an ECD change rate of <−6.1%/year). There were no significant differences in baseline characteristics between Groups A and B in terms of age, sex, laterality, follow-up years, or medical history of HTN, TB, DM, hepatitis, and CVA (Table 1). When tube parameters were compared, we found that TL was shorter, TID was shorter, and TCA was larger in Group A than in B (1.09 ± 0.37 mm vs. 1.27 ± 0.53 mm, *p* = 0.044; 0.23 ± 0.25 mm vs. 0.47 ± 0.46 mm, *p* = 0.004; 33.67 ± 11.24° vs. 28.73 ± 11.62°, *p* = 0.036, respectively; Table 2). In total, 51 out of 103 (50%) patients underwent hypertensive phase (HP), defined as IOP > 21 mmHg during the first 3 months after surgery, which was not associated with tube obstruction, retraction, or valve malfunction [22]. When IOP was above 21 mmHg during HP, glaucoma medications were added and we examined a significant decrease in final IOPs compared to preoperative IOPs (14.17 ± 4.51 vs. 26.02 ± 8.56 mmHg in overall patients, *p* < 0.001 from *t*-test).

### 3.3. Tube–Iris Distance Was the Only Statistically Significant Factor Predicting the ECD Change Rate

Univariable linear regressions evaluating the association between the ECD change rate and independent factors revealed three statistically significant factors (Table 3): the final ECD (β = 8.70 × 10^−5^ *p* < 0.001), the TL (β = −0.0422, *p* = 0.047), and the TID (β = −0.0722, *p* = 0.005). We included the latter two parameters in the multivariable regression analysis but excluded the final ECD, since final values cannot be used to predict survival. We also included the parameters found to be statistically significant in previous reports (TCD, TCA, and ITC). The correlation matrix between independent variables showed a high correlation between TCD and TCA (r = 0.78 via the Pearson method), which was above our predetermined cut-off value of 0.7 (Appendix A). It was difficult to identify which variable to include when comparing TCD and TCA because both showed moderate correlations with TL (r = 0.55) and TID (r = −0.54), respectively. Hence, multivariable analyses were performed twice after excluding either of them. In both regressions, TID was the only factor that showed statistical significance regarding the ECD change rate (β = −0.0684, *p* = 0.025 when TCD was excluded; β = −0.0743, *p* = 0.015 when TCA was excluded; Table 4).

### 3.4. Determination of the Optimal Cut-Off Value for TID

In our first approach, implemented to inspect the optimal cut-off values for TID, consecutive Cox proportional hazard analyses were conducted over every potential cut-off value between 0.01 mm and 1.25 mm at 0.01 mm intervals (Figure 2A). All cut-off values between 0.28 mm and 0.65 mm showed statistical significance regarding predicting survival (*p* < 0.05), while a TID of ≥0.33 mm showed the smallest *p*-value (*p* = 0.0087) and the largest impact (HR = 2.39). In our second approach, a univariable logistic regression model using TID as the independent variable showed an AUC of 0.662. An optimal cut-off value of 0.371 mm was determined by finding the point of contact of a line with a slope of 1 on the ROC curve. The sensitivity and specificity associated with this value were 0.553 and 0.769, respectively (Figure 2B).

### 3.5. Comparison between Patients with Short and Long TID

We also divided patients into two groups to draw KM curves based on TID values, with 0.33 mm and 0.371 mm as the respective cut-off values (Figure 3). The median survival times of patients with a TID of <0.33 mm, a TID of <0.371 mm, a TID of ≥0.33 mm, and a TID of ≥0.371 mm were 8.66 years (95% CI, 4.82–inf), 7.44 years (95% CI, 4.68–inf), 4.28 years (95% CI, 3.73–6.78), and 4.39 years (95% CI, 3.73–6.90), respectively. KM survival curves of short and long TID showed statistically significant differences regarding either cut-off value (cut-off, 0.33 mm, *p* = 0.00682 by the log-rank test; cut-off, 0.371 mm, *p* = 0.0459 by the log-rank test). Postoperative final IOP values were also compared via *t*-tests and did not show any differences between short and long TIDs (cut-off, 0.33 mm, 14.36 ± 4.01 mmHg vs. 13.83 ± 5.34 mmHg, *p* = 0.569; cut-off, 0.371 mm, 14.33 ± 4.03 mmHg vs. 13.84 ± 5.45 mmHg, *p* = 0.612; Appendix A).

We also thoroughly reviewed medical records to examine uveitis onset after AGV implantation. Of those not initially diagnosed with secondary glaucoma due to uveitis, only one patient had postoperative uveitis, with a TID of 0.05 mm (Appendix A). No significant differences in uveitis incidence between patients with short and long TID were identified (*p* = 1.00).

### 3.6. IID/TCA Was Significantly Associated with ECD Change Rate

We additionally conducted several univariable linear regressions to inspect associations between variables related to the tube–corneal angle or distance and the ECD change rate. Only IID/TCA was significantly associated with the ECD change rate, with a negative coefficient (β = −1.2467, *p* = 0.021, Appendix A).

## 4. Discussion

In our study the mean ECD change rate after tube insertion was −4.26 ± 9.49 (%/year), showing similar or less ECD change rate with some minimally invasive glaucoma surgeries. For instance, Oddone et al. conducted a study regarding XEN implants and reported a mean ECD reduction of −5.6% per year and Ibarz-Barberá et al. reported −7.4% ECD loss after PRESERFLO implantation [23,24].

However, our study adopted the hypothesis that not all glaucoma patients show ECD decline after AGV implantation [18]. The proportion of patients classified into the ECD reduction group was 36.9%, comparable to that reported by Lee et al. (32.3%). The mean ECD change rate in those patients without ECD reduction was 0.72 ± 5.42 (%/year), which is similar to that of patients receiving glaucoma medication without undergoing surgery reported by previous studies from South Korea (−3.7 ± 5.2%/year and −0.1 ± 2.4%/year) [6,9]. This supported our a priori hypothesis. In agreement with a previous study [16], univariable linear regression showed no significant associations between the ECD change rate and glaucoma type, including uveitic glaucoma.

Several variables (TCD, TCA, ITC) that had a statistically significant effect on ECD loss in previous studies were found to be non-significant. A low TCD was correlated with increased ECD loss in a multivariable analysis performed by Koo et al. [17]. However, collinearity between independent variables was not checked in that study, although this is encouraged before applying linear regression analyses, especially when the purpose is to obtain a lean and operationalized model [25]. In our study, TCD and TCA showed a high correlation (r = 0.78), which surpassed our prespecified cut-off of 0.7 (adopted from Donath et al.) [21]. Thus, either variable was excluded from our multivariable-adjusted models, which might account for the differences between the study by Koo et al. and this one. Different TCD distributions could also underlie these contrasting findings. Secondly, unlike a recent study by Lee et al. in which TCA was found to be a strong factor for predicting ECD loss, our study did not find the significance of TCA in linear regressions. Only the statistical significance of TCA in a comparative *t*-test evaluating differences between Groups A and B demonstrated the same tendency. This discrepancy might arise from differences in the sample distribution. Our population showed a wider TCA distribution, with a standard deviation twice as large as that detected by Lee et al. Lastly, a higher PAS was previously reported to correlate with a smaller central ECD [19]. Therefore, we subjected the ITC variable to multivariable linear regression, but the results were non-significant. This implies that the ITC might show significance in cross-sectional but not in time-series data, but this should be confirmed in future research.

We used two different methods to determine the optimal cut-off value for TID for predicting endothelial cell damage. Both methods provided a similar value and the log-rank tests using either of them were significant. The median survival time of the short TID group showed an infinite upper limit for the 95% confidence interval (CI). However, we believe that this finding was due to only a few patients showing large ECD losses in the short TID group, and therefore spurious. 

To the best of our knowledge, this study is the first to consider the AGV insertion site as a tube parameter. In a fixed ocular structure, the insertion–iris distance, tube–corneal angle, and tube length are the only variables necessary to calculate the distance from the tube tip to the iris. Therefore, we assert that the TID is an important endpoint variable that can be expressed as a function of eye structure, IID, TCA, and TL values. The fact that TCA does not show statistical significance regarding the ECD change rate can be understood in this context. With a fixed TCA, the TID increases or decreases with the IID. We investigated this issue by conducting univariable linear regressions using combinations of variables (Appendix A). AXL and ACW were included as representations of the eye structure. Only the IID/TCA ratio showed a significant correlation with the ECD change rate. A negative coefficient regarding the IID/TCA ratio implies that, for the purpose of ECD preservation, when the IID is fixed, it is better to have a larger TCA, and when the TCA is fixed, it is better to have a smaller IID. 

Our study has additional limitations. First, the sample enrolled comprised a relatively homogenous group of patients who were conservatively selected. However, the study was not pre-planned and selection bias could not be avoided due to its retrospective nature. Moreover, we did not include control groups but instead classified patients according to ECD change rates, which might have affected our result. This limitation was compensated by comparing the ratio of patient numbers and the mean ECD change rate in each group with findings from previous studies. Second, the frequencies and intervals of specular microscopic examinations differed among patients. Therefore, we calculated the slope of the ECD change over time using linear regression to compare the most appropriate ECD change rate between patients. Third, it has been reported that tube parameters in AS-OCT images can present differences over time [26]. Although we did not measure multiple AS-OCT images from each patient, further studies should consider this possibility. Lastly, we evaluated only the central region of the cornea for ECD calculation, but it has been shown that the peripheral area, especially the superotemporal area, is susceptible to greater ECD loss [6,17,18]. A plausible mechanism for this finding is that progenitor cells from the peripheral endothelium are more affected by jet flow or turbulence when the tube is inserted [27]. Additional prospective studies evaluating ECD in the entire cornea and considering the tube parameters evaluated here will provide more comprehensive insights.

Despite these limitations, the long follow-up duration, which is essential for observing ECD changes over time, is a substantial strength of our study. The mean follow-up time was 4.09 ± 2.20 years, which is higher than that from several previous studies (3 years, Tan et al.; 29.30 ± 14.67 months, Lee et al.; 2.5 ± 2.6 years, Koo et al.; 2 years, Lee et al.) [6,16,17,18].

## 5. Conclusions

In conclusion, our study suggests that TID is a feasible surrogate marker for surgeons to monitor during AGV implantations. We have shown that the TID can be determined as a function of the eye structure, the IID, the TCA, and the intracameral TL. The optimal cut-off value for the TID was found to be either 0.33 mm or 0.371 mm (depending on the methodology). Thus, our findings suggest that the tube should be inserted close enough to the iris to make the TID shorter than the cut-off value. Our findings guide future research directions and inform medical guidelines regarding surgical planning and follow-up protocols.

## Figures and Tables

**Figure 1 jcm-11-05057-f001:**
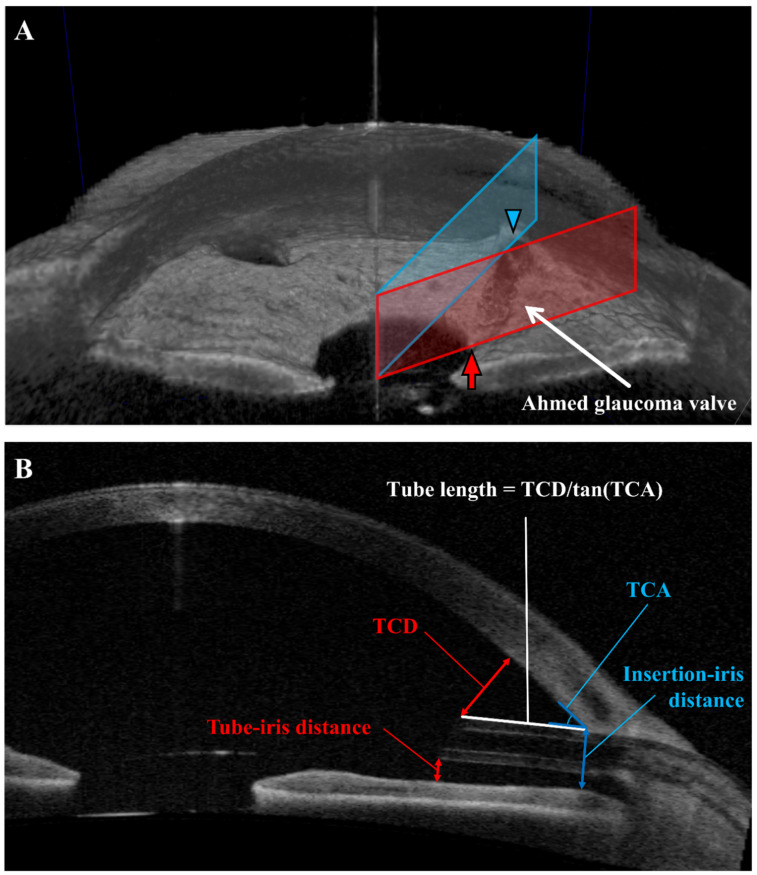
Measurement of tube parameters using anterior segment optical coherence tomography (AS-OCT). Two cross-sections either including depictions of the tube tip (red) or the tube insertion (blue), respectively, are shown in this 3D view. The red arrow indicates the tube tip, and the blue arrowhead shows the insertion site of the tube (**A**). Tube–corneal distance (TCD), tube–iris distance (TID), tube–corneal angle (TCA), and insertion–iris distance (IID) were measured directly, and tube length (TL) was computed as follows: TCD/tan (TCA). The cross-section in this figure was selected from a patient, wherein the tube tip and the insertion site could be seen in the same cross-section plane (**B**).

**Figure 2 jcm-11-05057-f002:**
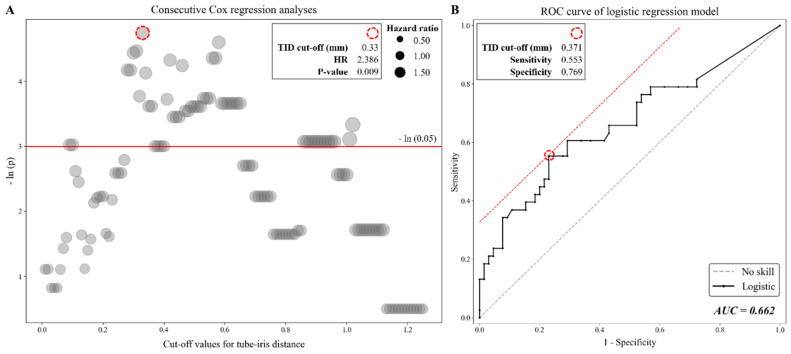
Determination of the optimal cut-off value regarding the tube–iris distance (TID). Consecutive Cox regression analyses were conducted for every potential cut-off value between 0.01 mm and 1.25 mm at 0.01 mm intervals. The x-axis indicates the evaluated TID cut-off values, the y-axis indicates the −ln (*p*) of the *p*-value for each cut-off (and is demarcated by a red horizontal line showing the value of −ln (0.05)), and the circle size of the determined data points mark the values of the associated hazard ratios (HR) (**A**). A receiver operating characteristic (ROC) curve was plotted using an univariable logistic regression model employing TID as the independent variable, and the area under the ROC curve (AUC) was calculated (**B**).

**Figure 3 jcm-11-05057-f003:**
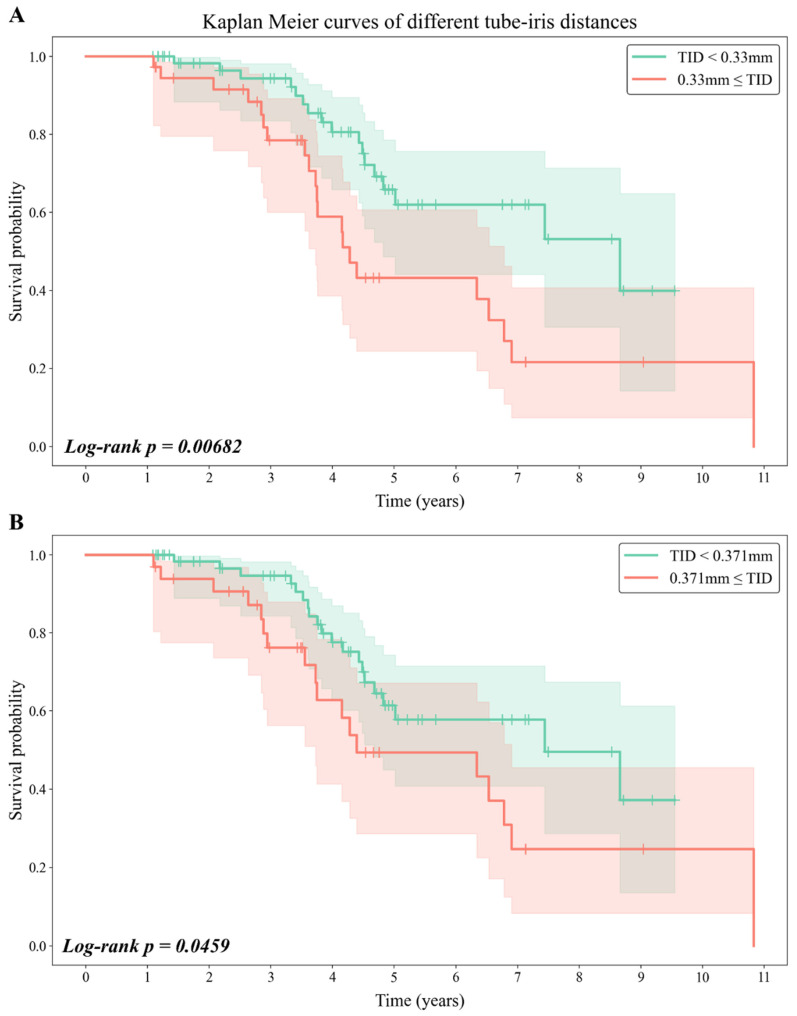
Kaplan–Meier curves of different tube–iris distances. Kaplan–Meier curves and log-rank tests were conducted for comparing patients with short or long tube–iris distances (TIDs), divided by the cut-off value determined from evaluations presented in Figure 2. Survival curves show significant difference with *p*-value of 0.00682 when 0.33 mm is used as the cut-off (**A**). When 0.371 mm is used as the cut-off, survival curves also show significant difference with *p*-value of 0.0459 (**B**).

**Table 1 jcm-11-05057-t001:** Patient medical and demographic characteristics (n = 103).

Baseline	Overall	Patients without an ECD Decline (Group A, n = 65)	Patients with an ECD Decline (Group B, n = 38)	*p*-Value
Age	64.20 ± 13.85	63.95 ± 13.79	64.63 ± 14.12	0.812
Sex (M/F)	64/39	40/25	24/14	0.963 *
Laterality (R/L)	56/47	38/27	18/20	0.376 *
AXL (mm)	24.80 ± 1.89	24.78 ± 1.86	24.82 ± 1.96	0.919
CCT (μm)	521.37 ± 51.27	518.60 ± 51.91	526.11 ± 50.49	0.476
Follow-up (years)	4.09 ± 2.20	4.02 ± 2.32	4.22 ± 2.01	0.663
Preoperative IOP (mmHg)	26.02 ± 8.56	25.56 ± 8.68	26.82 ± 8.41	0.475
Final IOP (mmHg)	14.17 ± 4.51	14.68 ± 4.56	13.30 ± 4.36	0.135
Preoperative ECD (cells/mm^2^)	2183.42 ± 527.69	2215.87 ± 473.24	2127.90 ± 612.57	0.417
Final ECD (cells/mm^2^)	1765.49 ± 663.50	2119.09 ± 494.94	1160.66 ± 441.07	<0.001
ECD change rate (%/year)	−4.26 ± 9.49	0.72 ± 5.42	−12.78 ± 8.9	<0.001
**Systemic disease**				
Hypertension	43 (41.75)	30 (46.15)	13 (34.21)	0.328 *
Tuberculosis	5 (4.85)	3 (4.62)	2 (5.26)	1.000 †
Diabetes mellitus	38 (36.89)	29 (44.62)	9 (23.68)	0.056 *
Hepatitis	2 (1.94)	2 (3.08)	0 (0.0)	0.530 †
Cerebrovascular accident	5 (4.85)	4 (6.15)	1 (2.63)	0.649 †
**Glaucoma**				
POAG	57 (55.34)	37 (56.92)	20 (52.63)	0.828 *
Chronic angle-closure glaucoma	3 (2.91)	2 (3.08)	1 (2.63)	1.000 †
Neovascular glaucoma	11 (10.68)	8 (12.31)	3 (7.89)	0.742 †
Pigmentary glaucoma	1 (0.97)	1 (1.54)	0 (0.0)	1.000 †
Pseudoexfoliation glaucoma	3 (2.91)	2 (3.08)	1 (2.63)	1.000 †
Secondary (d/t uveitis)	12 (11.65)	6 (9.23)	6 (15.79)	0.495 *
Secondary other	16 (15.53)	9 (13.85)	7 (18.42)	0.736 *

AXL, axial length; CCT, central corneal thickness; IOP, intraocular pressure; ECD, endothelial cell density; POAG, primary open angle glaucoma. * χ2 test; † Fisher’s exact test; otherwise *t*-test.

**Table 2 jcm-11-05057-t002:** Comparison of tube parameters between the patients enrolled in Group A and Group B.

Tube Parameters	Overall	Patients without an ECD Decline (Group A, n = 65)	Patients with an ECD Decline (Group B, n = 38)	*p*-Value
Tube length (mm)	1.15 ± 0.44	1.09 ± 0.37	1.27 ± 0.53	0.044 *
Tube–corneal distance (mm)	0.77 ± 0.47	0.77 ± 0.40	0.76 ± 0.57	0.932
Tube–iris distance (mm)	0.32 ± 0.36	0.23 ± 0.25	0.47 ± 0.46	0.004 *
Tube–corneal angle (°)	31.85 ± 11.58	33.67 ± 11.24	28.73 ± 11.62	0.036 *
Insertion–iris distance (mm)	0.68 ± 0.27	0.66 ± 0.24	0.71 ± 0.32	0.372
Tube location (ST/SN) †	101/2	63/2	38/0	0.530

ECD, endothelial cell density; † Fisher’s exact test; * *p* < 0.05 from *t*-test.

**Table 3 jcm-11-05057-t003:** Univariable linear regressions regarding associations with ECD change rates.

Variables	Univariable Analysis		
	β	95% Confidence Interval	*p*-Value
Age (years)	1.10 × 10^−5^	[−1.34 × 10^−3^, 1.36 × 10^−3^]	0.987
Sex (M/F)	−4.30 × 10^−3^	[−4.27 × 10^−2^, 3.41 × 10^−2^]	0.825
Laterality (R/L)	3.40 × 10^−5^	[−3.74 × 10^−2^, 3.74 × 10^−2^]	0.999
AXL (mm)	1.70 × 10^−4^	[−9.74 × 10^−3^, 1.01 × 10^−2^]	0.973
CCT (μm)	−7.70 × 10^−5^	[−4.42 × 10^−4^, 2.88 × 10^−4^]	0.676
Follow-up (years)	−3.26 × 10^−3^	[−1.17 × 10^−2^, 5.21 × 10^−3^]	0.447
Preoperative IOP (mmHg)	−2.52 × 10^−4^	[−2.44 × 10^−3^, 1.93 × 10^−3^]	0.819
Final IOP (mmHg)	2.89 × 10^−3^	[−1.22 × 10^−3^, 7.00 × 10^−3^]	0.166
Preoperative ECD(cells/mm^2^)	−2.10 × 10^−5^	[−5.70 × 10^−5^, 1.40 × 10^−5^]	0.232
Final ECD (cells/mm^2^)	8.70 × 10^−5^	[6.40 × 10^−5^, 1.09 × 10^−4^]	<0.001 *
**Systemic disease**			
Hypertension	2.01 × 10^−2^	[−1.75 × 10^−2^, 5.77 × 10^−2^]	0.292
Tuberculosis	−6.50 × 10^−3^	[−9.32 × 10^−2^, 8.02 × 10^−2^]	0.882
Diabetes mellitus	2.64 × 10^−2^	[−1.19 × 10^−2^, 6.47 × 10^−2^]	0.174
Hepatitis	9.97 × 10^−2^	[−3.39 × 10^−2^, 2.33 × 10^−1^]	0.142
Cerebrovascular accident	5.60 × 10^−2^	[−3.00 × 10^−2^, 1.42 × 10^−1^]	0.199
**Glaucoma**			
POAG	7.25 × 10^−3^	[−3.02 × 10^−2^, 4.47 × 10^−2^]	0.702
Chronic angle-closure glaucoma	9.86 × 10^−3^	[−1.01 × 10^−1^, 1.21 × 10^−1^]	0.860
Neovascular glaucoma	2.99 × 10^−2^	[−3.02 × 10^−2^, 8.99 × 10^−2^]	0.326
Pigmentary glaucoma	7.35 × 10^−2^	[−1.16 × 10^−1^, 2.63 × 10^−1^]	0.443
Pseudoexfoliation glaucoma	−3.57 × 10^−2^	[−1.46 × 10^−1^, 7.49 × 10^−2^]	0.524
Secondary (d/t uveitis)	−2.87 × 10^−2^	[−8.65 × 10^−2^, 2.91 × 10^−2^]	0.326
Secondary other	−1.27 × 10^−2^	[−6.40 × 10^−2^, 3.87 × 10^−2^]	0.626
**Tube parameters**			
Tube length (mm)	−4.22 × 10^−2^	[−8.38 × 10^−2^, −5.64 × 10^−4^]	0.047 *
Tube–corneal distance (mm)	−4.04 × 10^−3^	[−4.41 × 10^−2^, 3.60 × 10^−2^]	0.842
Tube–iris distance (mm)	−7.22 × 10^−2^	[−1.22 × 10^−1^, −2.23 × 10^−2^]	0.005 *
Tube–corneal angle (°)	1.11 × 10^−3^	[−4.94 × 10^−4^, 2.71 × 10^−3^]	0.173
Insertion–iris distance (mm)	−6.73 × 10^−2^	[−1.35 × 10^−1^, 5.87 × 10^−4^]	0.052
Tube location (ST/SN)	−6.53 × 10^−2^	[−2.00 × 10^−1^, 6.91 × 10^−2^]	0.337
**Anterior chamber parameters**			
ACD (mm)	−2.37 × 10^−2^	[−4.87 × 10^−2^, 1.33 × 10^−3^]	0.063
ITC (%)	−9.30 × 10^−5^	[−7.19 × 10^−4^, 5.34 × 10^−4^]	0.770
ACW (mm)	1.82 × 10^−2^	[−2.27 × 10^−2^, 5.91 × 10^−2^]	0.379
nasal AOD500 (μm)	3.00 × 10^−5^	[−4.10 × 10^−5^, 1.01 × 10^−4^]	0.405
nasal AOD750 (μm)	1.20 × 10^−5^	[−4.10 × 10^−5^, 6.60 × 10^−5^]	0.650
nasal ARA500 (μm)	4.70 × 10^−5^	[−1.20 × 10^−4^, 2.15 × 10^−4^]	0.576
nasal ARA750 (μm)	2.90 × 10^−5^	[−7.20 × 10^−5^, 1.30 × 10^−4^]	0.575
nasal TISA500 (μm)	7.00 × 10^−5^	[−1.18 × 10^−4^, 2.59 × 10^−4^]	0.461
nasal TISA750 (μm)	3.60 × 10^−5^	[−7.20 × 10^−5^, 1.44 × 10^−4^]	0.510
nasal TIA500 (°)	3.25 × 10^−4^	[−7.94 × 10^−4^, 1.45 × 10^−3^]	0.565
nasal TIA750 (°)	2.26 × 10^−4^	[−1.02 × 10^−3^, 1.47 × 10^−3^]	0.719
temporal AOD500 (μm)	3.10 × 10^−5^	[−3.50 × 10^−5^, 9.80 × 10^−5^]	0.350
temporal AOD750 (μm)	1.50 × 10^−5^	[−3.40 × 10^−5^, 6.50 × 10^−5^]	0.540
temporal ARA500 (μm)	9.70 × 10^−5^	[−5.00 × 10^−5^, 2.43 × 10^−4^]	0.194
temporal ARA750 (μm)	4.60 × 10^−5^	[−4.50 × 10^−5^, 1.37 × 10^−4^]	0.317
temporal TISA500 (μm)	9.50 × 10^−5^	[−7.30 × 10^−5^, 2.62 × 10^−4^]	0.264
temp_TISA_750 (μm)	4.70 × 10^−5^	[−5.00 × 10^−5^, 1.45 × 10^−4^]	0.337
temp_TIA_500 (°)	3.73 × 10^−4^	[−6.68 × 10^−4^, 1.41 × 10^−3^]	0.479
temp_TIA_750 (°)	4.02 × 10^−4^	[−7.94 × 10^−4^, 1.60 × 10^−3^]	0.506

AXL axial length, CCT central corneal thickness, IOP intraocular pressure, ECD endothelial cell density, POAG primary open angle glaucoma, ACD anterior chamber depth, ITC iris trabecular meshwork contact, ACW anterior chamber width, AOD angle opening distance, ARA angle recess area, TISA trabecular iris space area, TIA trabecular iris angle, * *p* < 0.05

**Table 4 jcm-11-05057-t004:** Multivariable linear regressions regarding associations with endothelial cell density change rates. Due to the high collinearity between TCD and TCA, one or the other of these variables was excluded when conducting multivariable analyses.

Variables	Multivariable Analysis (TCD Excluded)			Multivariable Analysis (TCA Excluded)		
	β	95% CI	*p*-Value	β	95% CI	*p*-Value
**Tube parameters**						
Tube length (mm)	−3.71 × 10^−2^	[−7.80 × 10^−2^, 4.00 × 10^−3^]	0.076	−3.00 × 10^−2^	[−8.40 × 10^−2^, 2.40 × 10^−2^]	0.269
Tube–corneal distance (mm)				−1.13 × 10^−2^	[−6.60 × 10^−2^, 4.40 × 10^−2^]	0.683
Tube–iris distance (mm)	−6.84 × 10^−2^	[−1.28 × 10^−1^, −9.00 × 10^−3^]	0.025 *	−7.43 × 10^−2^	[−1.34 × 10^−1^, −1.50 × 10^−2^]	0.015 *
Tube–corneal angle (°)	−3.52 × 10^−5^	[−2.00 × 10^−3^, 2.00 × 10^−3^]	0.970			
**Anterior chamber parameters**						
ITC (%)	−6.25 × 10^−5^	[−1.00 × 10^−3^, 1.00 × 10^−3^]	0.838	−7.03 × 10^−5^	[−1.00 × 10^−3^, 1.00 × 10^−3^]	0.818

CI, confidence interval; ITC, iris trabecular meshwork contact; * *p* < 0.05.

## Data Availability

The datasets used and/or analyzed during the current study are available from the corresponding author upon reasonable request.

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
