# Peer review of "Tube–Iris Distance and Corneal Endothelial Cell Damage Following Ahmed Glaucoma Valve Implantation"

_jcm, 2022, doi:10.3390/jcm11175057_

Round 1
Reviewer 1 Report
Dear Authors,
The paper titled ‘Tube-iris Distance and Corneal Endothelial Cell Damage following Ahmed Glaucoma Valve Implantation’ is a well-written article about an important topic.
I have some questions regarding this article.
1. Was IOP control achieved with or without medications after tube implantation?
2. What was the primary cause in NVG cases? Have these patients received any additional treatment?
3. After tube surgery did the hypertensive phase develop in patients?
4. During the surgery, did they use viscoelastic materials to maintain anterior chamber depth?
Author Response
The paper titled ‘Tube-iris Distance and Corneal Endothelial Cell Damage following Ahmed Glaucoma Valve Implantation’ is a well-written article about an important topic.
I have some questions regarding this article.
Response: We appreciate the reviewer’s comments regarding our manuscript. We revised the manuscript as suggested by the reviewer, as shown below.
Comments:
- Reviewer’s comment: Was IOP control achieved with or without medications after tube implantation?
Response: Thank you for a valuable comment. When IOP was above 21mmHg during hypertensive phase, glaucoma medications were added. 66 out of 103 (64%) patients underwent hypertensive phase (HP), defined as IOP > 21 mmHg during the first 3 months after surgery which was not associated with tube ob-struction, retraction, or valve malfunction. We investigated the distribution of final post-operative IOPs and compared it with the distribution of pre-operative IOPs using t-test. The result was as: 14.17 ± 4.51 vs. 26.02 ± 8.56mmHg in overall patients. (p < 0.001) we have added this in the manuscript as:
66 out of 103 (64%) patients underwent hypertensive phase (HP), defined as IOP > 21 mmHg during the first 3 months after surgery which was not associated with tube obstruction, retraction, or valve malfunction. When IOP was above 21mmHg during HP, glaucoma medications were added and we examined a significant decrease of final IOPs compared to preoperative IOPs (14.17 ± 4.51 vs. 26.02 ± 8.56mmHg in overall patients, p < 0.001 from t-test) (Page 6, lines 221)
- Reviewer’s comment: What was the primary cause in NVG cases? Have these patients received any additional treatment?
Response: Among 11 cases of NVG, diabetic retinopathy was the primary cause (four proliferative diabetic retinopathies and four non-proliferative diabetic retinopathies), while other three cases were caused by retinal vein occlusions. Additional treatments include panretinal photocoagulation and intravitreal anti VEGF injections. We added the primary causes of eyes with NVG cases and noted the additional treatment implemented in these patients as:
Among 11 cases of NVG, diabetic retinopathy was the primary cause (four proliferative diabetic retinopathies and four non-proliferative diabetic retinopathies), while other three cases were caused by retinal vein occlusions. Additional treatments include panretinal photocoagulation and intravitreal anti VEGF injections. (Page 5, lines 203)
- Reviewer’s comment: After tube surgery did the hypertensive phase develop in patients?
Response: We thank the reviewer for pointing this out. Hypertensive phase after surgery is an important aspect to discuss in glaucomatous eyes after surgery. 66 out of 103 (64%) patients underwent hypertensive phase (HP), defined as IOP > 21 mmHg during the first 3 months after surgery which was not associated with tube obstruction, retraction, or valve malfunction We added the ratio of patients in which hypertensive phase was developed as:
66 out of 103 (64%) patients underwent hypertensive phase (HP), defined as IOP > 21 mmHg during the first 3 months after surgery which was not associated with tube obstruction, retraction, or valve malfunction (Page 6, lines 221)
- Reviewer’s comment: During the surgery, did they use viscoelastic materials to maintain anterior chamber depth?
Response: Thank you for a valuable comment. AGV implantation surgeries were done using viscoelastic materials, and we added this information in the method section as:
Viscoelastic was injected to maintain the anterior chamber depth before tube insertion. (Page 3, line 101)
Reviewer 2 Report
Thank you for the opportunity to review the paper by Yitak Kim and colleagues.
My general comment is that the study is well done, the topic of interest and the results relevant.
My minor suggestions:
language smoothing
the paper is overall long, I suggest where is possible to shorten the manuscript
table 3 is crowded and difficult to follow, I suggest to simplify or split in two tabs.
although the paper is about tubes, as comparison consider to mention EC loss after MIGS (xen: Endothelial Cell Density After XEN Implant Surgery: Short-term Data From the Italian XEN Glaucoma Treatment Registry (XEN-GTR). Oddone F et al. - preserflo: Corneal Endothelial Cell Loss After PRESERFLO MicroShunt Implantation in the Anterior Chamber: Anterior Segment OCT Tube Location as a Risk Factor. Marta Ibarz-Barbera ́et al.)
Author Response
Thank you for the opportunity to review the paper by Yitak Kim and colleagues. My general comment is that the study is well done, the topic of interest and the results relevant.
Response: We appreciate the reviewer’s comments regarding our manuscript. We revised the manuscript as suggested by the reviewer, as shown below.
Comments:
- Reviewer’s comment: Table 3 is crowded and difficult to follow, I suggest to simplify or split in two tabs.
Response: Thank you for a valuable comment. We do agree with the reviewer’s opinion and split Table 3 into two parts, according to whether it was an univariable analysis or a multivariable analysis.
- Reviewer’s comment: Although the paper is about tubes, as comparison consider to mention EC loss after MIGS (xen: Endothelial Cell Density After XEN Implant Surgery: Short-term Data From the Italian XEN Glaucoma Treatment Registry (XEN-GTR). Oddone F et al. - preserflo: Corneal Endothelial Cell Loss After PRESERFLO MicroShunt Implantation in the Anterior Chamber: Anterior Segment OCT Tube Location as a Risk Factor. Marta Ibarz-Barbera ́et al.)
Response: We appreciate the reviewer’s comment. We compared the rate of ECD loss from our study to the above-mentioned methodologies and added it in the discussion section as following.
In our study the mean ECD change rate after tube insertion was - 4.26 ± 9.49 (%/year). Showing similar or less ECD change rate with some minimally-invasive glaucoma surgeries. For instance, Oddone et al. conducted a study regarding XEN implants and reported a mean ECD reduction of -5.6% per year and Ibarz-Barberá et alreported -7.4% ECD loss after PRESERFLO implantation. [23,24] (page 11, line 305)